# Relationship Between Anthropometric Profile, Body Composition, and Physical Performance in Spanish Professional Female Soccer Players at Pre-Season Onset: A Cross-Sectional Study

**DOI:** 10.3390/jfmk10010079

**Published:** 2025-02-27

**Authors:** Marta Ramírez-Munera, Raúl Arcusa, Francisco Javier López-Román, Vicente Ávila-Gandía, Silvia Pérez-Piñero, Juan Carlos Muñoz-Carrillo, Antonio Jesús Luque-Rubia, Javier Marhuenda

**Affiliations:** 1Faculty of Pharmacy and Nutrition, Campus de los Jerónimos, Guadalupe, Universidad Católica de Murcia (UCAM), 30107 Murcia, Spain; marta.ramirez.nutricionista@gmail.com (M.R.-M.); jmarhuenda@ucam.edu (J.M.); 2Faculty of Medicine, Campus de los Jerónimos, Guadalupe, Universidad Católica de Murcia (UCAM), 30107 Murcia, Spain; jlroman@ucam.edu (F.J.L.-R.); vavila@ucam.edu (V.Á.-G.); sperez2@ucam.edu (S.P.-P.); jcmunoz@ucam.edu (J.C.M.-C.); ajluque@ucam.edu (A.J.L.-R.); 3Primary Care Research Group, Biomedical Research Institute of Murcia (IMIB-Arrixaca), 30120 Murcia, Spain

**Keywords:** anthropometry, physical activity, athletic training, sports performance, morphology

## Abstract

**Background**: Body composition plays a key role in both anaerobic and aerobic performance in professional soccer. However, its relationship with physical performance in female players remains underexplored. Assessing these parameters at the beginning of the pre-season provides valuable insights for optimizing player readiness. This study investigates the correlation between anthropometric profile, body composition, and physical performance in professional female soccer players at pre-season onset. Additionally, it analyzes positional differences and their relation to the specific physical demands of women’s soccer. **Methods**: A cross-sectional study of thirty-four female soccer players (age: 23.06 ± 4.29 years, height: 164.15 ± 5.84 cm, weight: 58.39 ± 6.62 kg, and ∑6 skinfolds: 74.57 ± 18.48 mm) who performed anthropometric measurements, the Countermovement Jump (CMJ), Wingate Anaerobic Test (WAnT), and Yo-Yo Intermittent Recovery Level 1 (Yo-Yo IR1) on the first day of the pre-season. One-way ANOVA with Tukey’s post hoc test assessed positional differences, while Pearson’s or Spearman’s correlation coefficients evaluated relationships between anthropometric variables and performance outcomes. **Results**: Very high positive correlations were found between body and muscle mass with the power variables in CMJ and WAnT (r = 0.70–0.89; *p* < 0.001). An almost perfect correlation was observed between body mass and peak power in WAnT (r = 0.904; *p* < 0.001). In contrast, fat mass showed moderate negative correlations with jump height and aerobic performance (Yo-Yo IR1 distance) (r = 0.30–0.49; *p* < 0.04). Significant differences were observed in the mean (757.60 ± 95.59 W) and peak power (773.59 ± 101.88 W) of CMJ pre-WAnT, with goalkeepers showing higher values compared to defenders and midfielders (*p* < 0.05). **Conclusions**: Body composition significantly influences athletic performance, showing positive correlations of muscle and body mass with anaerobic power and capacity. In contrast, fat mass showed negative correlations with explosive strength, recovery, and aerobic capacity.

## 1. Introduction

The development of women’s soccer, both on and off the pitch, is progressing at an unprecedented rate. Key drivers of this growth include recent reforms in competition structures, the modernization of player development programs, and the continued professionalization of the women’s game [1]. These factors collectively contribute to enhancing performance, increasing visibility, and promoting gender equity within the sport [2]. However, research on women’s soccer remains comparatively limited, particularly compared with male players [3].

Soccer is a multidirectional, high-intensity intermittent sport in which success depends on the physical, technical, tactical, and body composition aspects of players [4,5,6,7,8]. Anthropometry provides a systematic method for measuring, quantifying, and characterization of the morphological composition of the human body [9]. Morphological characteristics and body composition have become critical factors, as they significantly influence performance in soccer [8,10,11].

From a physiological perspective, soccer can be characterized as predominantly aerobic exercise combined with frequent, intermittent, short, intense actions with a high rate of anaerobic energy turnover [12]. Players’ recovery capacity during repeated high-intensity efforts is strongly related to the development of aerobic capacity [13,14], often assessed using the Yo-Yo IR1 [15]. On the other hand, explosive-type efforts, such as sprints or jumps, are important factors for soccer performance [16]. These efforts depend on the anaerobic capacity of the neuromuscular system, specifically on the explosive strength and anaerobic power of lower limbs [17], commonly evaluated using tests such as the CMJ [18,19,20] and the WAnT [21].

The relationship between anthropometric characteristics and physical performance can assist in evaluating the specificity of pre-season training programs. Moreover, it is essential to consider that in high-performance sports contexts such as soccer, the pre-season is often relatively short compared to the competitive period. This makes it increasingly important to assess and monitor body composition and physical fitness parameters before the competitive period begins in order to have a more complete assessment of the players’ initial state [22]. However, information on physical performance metrics at the beginning of the pre-season, especially right after the off-season, is also lacking. To our knowledge, this is the first study to explore the relationship between anthropometric profiles, body composition, and performance using Yo-Yo IR1, CMJ, and WAnT on the first days of the preseason.

The primary aim of this study is to examine the correlation between anthropometric profile, body composition, and physical performance in professional female soccer players at the onset of the pre-season. A secondary aim is to examine how positional differences relate to the physical demands specific to women’s soccer.

We hypothesize that anthropometric profile and body composition play a significant role in the physical performance of female soccer players at the onset of the pre-season, with muscle mass having a positive impact on anaerobic performance and fat mass being negatively associated with key performance metrics.

## 2. Materials and Methods

### 2.1. Study Design

A cross-sectional study was carried out at the beginning of the pre-season. Recruitment and initial contact with the teams were conducted two months before the study, with player recruitment finalized one month prior to data collection. All determinations were performed on all players on the first day of the pre-season. On this day, anthropometric measurements, body composition measurements, and physical performance tests, including the CMJ and WAnT, were conducted at the facilities of the Catholic University San Antonio of Murcia (UCAM). Two days later, the Yo-Yo IR1 was performed at the respective training fields of the participating teams. Prior to starting the intervention, the protocol was approved by the Institutional Review Committee of the UCAM (code CE052204), following the Standards of Good Clinical Practice, and was conducted according to the Declaration of Helsinki. The clinical trial was registered at www.clinicaltrials.gov (accessed on 9 August 2022) (identifier NCT05525871). Current European legislation on the protection of personal data (Regulation (EU)2016/679) was complied with.

### 2.2. Participants

Research was conducted on a sample of 34 professional female players (age: 23.06 ± 4.29 years) from two soccer teams in the first and second Spanish professional Women’s Football League.

The inclusion criteria for participant recruitment were as follows: (a) healthy subject with medical authorization for the practice of federated sport; (b) belongs to a team of the first or second division of the Spanish Women’s League. The exclusion criteria for the study were: (a) changing teams during pre-season; (b) history of drug, alcohol, or other substance abuse or other factors limiting their ability to cooperate during the study. All players were previously informed of the objectives and method of the research, signing informed consent forms before starting the research. In the case of minor players, it was their parents who signed the consent.

### 2.3. Anthropometric Measurements

All anthropometric measurements, including basic measurements, eight skinfolds, six breadths, and six perimeters, were performed twice to minimize errors. These measurements were carried out by a level 2 anthropometrist, accredited by the International Society for the Advancement of Kinanthropometry (ISAK) [23], following the procedures established by that organization. The methodology and equipment used are detailed in a previous publication [24].

### 2.4. Countermovement Jump

Explosive strength, anaerobic power, and neuromuscular fatigue in the lower limbs were measured and analyzed using the CMJ [18,19,20] performed on a contact platform (Chronojump Boscosystem, Barcelona, Spain). The CMJ analysis was performed using Chronojump software, version 2.2.1. For correct execution, subjects placed their hands on their hips, bent their knees to a 90-degree angle, and jumped to maximum height in a single motion. The knee’s right angle was monitored in the sagittal plane using real-time video digitization software. The system included an A2-sized contact platform with two isolated electrical contacts acting as a pressure-dependent switch. This switch closed as the player stood on the platform. A PC connected to the platform calculated flight time with a temporal resolution of 1 m/s. The change in center of gravity during flight (jump height, h) was calculated using the measured flight time (t) with the standardized Equation (1):h (cm) = (t^2^ × g)/8(1)
where g is the acceleration due to gravity (9.81 m/s^2^). A total of three jumps were performed, with the average height recorded. This same protocol was applied for the CMJs performed before and after the WAnT (see Figure 1). From each CMJ test, jump height (cm), mean (CMJ_Pmean_), and peak power (W) produced (CMJ_Ppeak_) were extracted, as indicators of explosive strength, anaerobic power, and neuromuscular fatigue [25].

### 2.5. Wingate Anaerobic Test

Anaerobic power and anaerobic capacity were analyzed using the WAnT [21]. For the execution of this test, the optimal seat height on a Monark 894E weight ergometer (Monark, Vansbro, Sweden) was adjusted for the players. The seat height was set so that knee flexion was no more than 5 degrees with the leg fully extended. Each subject underwent a 3-min warm-up period at 60 revolutions per minute. Two all-out sprints of five seconds were included during the warm-up: the first sprint at one minute and thirty seconds, and the second sprint at two minutes and fifty-five seconds. This was done to familiarize them with the test. After the warm-up, subjects rested passively for two minutes before beginning the WAnT. The resistance load was set at 7.5% of the subject’s body weight [26]. The test procedure consisted of a 10-s countdown, followed by a 30-s all-out pedaling phase and an active recovery phase. During the first 5 s of the countdown, subjects remained stationary, with the dominant leg positioned forward in preparation for the start and the pedals slightly offset from being fully horizontal. At the end of the countdown, the resistance load was released, and subjects began pedaling. All subjects were verbally encouraged to continue pedaling as fast as they could for the 30 s. Power for the following 30 s was recorded using a video camera, and later, data as peak power (P_peak_), time (s) to reach P_peak_ (TP_peak_), mean power during the 30 s sprint (P_mean_), and minimal power (P_min_) were manually transcribed. The Fatigue Index (%) (FI) was calculated. In addition to P_mean_ during the entire sprint, mean power was also calculated every 10 s and 15 s of the sprint (Split_0–10S_, Split_10–20S_, Split_20–30S_, Split_0–15S_, Split_15–30S_).

### 2.6. Yo-Yo Intermittent Recovery (Level 1) Test

Aerobic capacity was analyzed, and maximal oxygen uptake (VO_2max_) was estimated using the Yo-Yo IR1 [15]. This test consists of 2 × 20 m shuttle runs at increasing speeds, interspersed with a 10-s period of active recovery (controlled by audio signals from a CD player). An individual is running until he/she is not able to maintain the speed. Finally, at the end of the test, the speed level, speed (km/h), time (s), and distance (m) were recorded. Additionally, VO_2max_ values of the players were estimated by using the following Equation (2) [15]:VO_2max_ (mL/min/Kg) = (Yo-Yo IR1 distance (m) × 0.0084 + 36.4)(2)

### 2.7. Statistical Analyses

SPSS Statistics 27 (SPSS, Inc., Chicago, IL, USA) was used for statistical analysis. Descriptive statistics were calculated and used to describe the physical performance variables of the players.

The statistics were reported as mean ± standard deviation (SD). The Shapiro–Wilk test was used to check the normal distribution of data. One-way analysis of variance (ANOVA) was used to determine differences between physical performance variables among the four playing positions. Tukey’s post hoc test was used to determine which variables differed significantly. The level of statistical significance was set at *p* ≤ 0.05. Pearson’s or Spearman’s correlation coefficient (*r*) was used to determine possible the correlation between anthropometric profile and body composition and physical performance variables.

## 3. Results and Discussion

The demographic characteristics of the players can be seen in Table 1. The anthropometric measurements and body composition of the players are available in detail in a previous publication by our research team [24].

The descriptive statistics for CMJ pre-WAnT are available in Table 2, including jump height, P_mean_, and P_peak_. Regarding jump height, goalkeepers showed the highest values. That fact agrees with the observations reported on Tunisian first-division players [27]. However, other studies reported higher jump heights for forwards [28,29,30,31] or defenders [32,33]. The greater jump height observed in goalkeepers may be partially attributed to their role-specific demands, as they regularly perform vertical and lateral jumps during both training and matches [27].

Despite these descriptive observations, no statistically significant differences were found between playing positions for jump height, confirming the findings in other studies [28,29,30,32,33]. However, some research has reported positional differences. For example, a study on Tunisian first-division players [27] reported that goalkeepers and forwards exhibited higher jump heights. Similarly, a study on the Chilean national team [31] found that defenders and forwards had higher values compared to other positions but did not include goalkeepers. It is important to highlight that there is a limited number of studies reporting the jump height of female soccer players in different playing positions, especially those including goalkeepers [27,28,30,32,33].

Following on from jump height and analyzing the lowest values, midfielders showed the lowest jump height. This finding is concordant with observations in Tunisian [27], Norwegian [33], and Serbian [29] first-division players and the national team of Chile [30,31]. Other studies reported the lower jump height in semi-professional South African goalkeepers [28] and Montenegrin national team forwards [32].

The jump height reported in the present research for all players (26.01 ± 4.42) is comparable to that observed in a previous study on Spanish first-division players in 2009 that measured the players during the season (26.10 ± 4.8) [16]. Since the measurements were obtained on the first day of the pre-season, this limits the comparison with other studies due to possible seasonal variations.

This research shows higher jump heights compared to pre-season data for lower-level Spanish players [34,35,36] and elite Serbian players [37]. However, these values are lower than those recorded in pre-season for Norwegian first-division players (32.6 ± 4.1) [33] and the Finnish national team (32.3 ± 3.4) [38]. Considering measurements obtained at different times of the season, the jump height observed was higher than those at the end of the season for the Montenegrin national team [32]. In turn, it was lower than those found in Danish [39], Spanish [40], Brazilian [8], Swedish [41], English [42], and Norwegian [41,43] first-division players, as well as the Italian [44] and Norwegian [43] national teams.

As observed in jump height, goalkeepers showed the highest CMJ_Pmean_ and CMJ_Ppeak_ values. However, in contrast to jump height, both variables were statistically different between positions (CMJ_Pmean_ (η^2^ partial = 0.252, *p* = 0.046) and CMJ_Ppeak_ (η^2^ partial = 0.249, *p* = 0.048). Specifically, goalkeepers had higher CMJ_Pmean_ compared to defenders (*p* = 0.056) and midfielders (*p* = 0.055) and higher CMJ_Ppeak_ compared to defenders (*p* = 0.058).

Following the examination of positional differences, the focus then shifts to understanding the correlations between CMJ performance variables and anthropometric and body composition measurements (Table 3). Initially, only those that were high, very high, or almost perfect—defined as having a correlation coefficient value greater than or equal to 0.5 [45] and a *p* < 0.001—were included. However, this criterion excluded moderate correlations between anthropometric variables and CMJ height, as well as Yo-Yo IR1 variables such as distance and VO_2max_. Thus, moderate correlations for these specific variables were included to provide a more comprehensive analysis.

Moreover, jump height, used as a measure of explosive strength, showed a moderate negative correlation with anthropometric variables related to fat mass, such as biceps skinfold, percentage, and kilograms of fat mass calculated using different equations, sum of six and eight skinfolds, and endomorphy. Additionally, a moderate positive correlation was found with muscle mass percentage (using Poortmans’ equation). These findings show a negative correlation between jump height and fat mass and a positive correlation with muscle mass. Those differences were consistent with studies on Serbian elite [37] and adolescent [46] players and the Chilean national team [30].

In turn, CMJ_Pmean_ and CMJ_Ppeak_ are used for the measurement of anaerobic power. They both showed a very high positive correlation with body mass and kilograms of muscle mass. This can be explained by the fact that power is the product of force and velocity (Power = Force × Velocity) and force is directly proportional to mass and acceleration (Force = Mass × Acceleration). As a result, players with greater body and muscle mass generate more force, which leads to higher power output.

Similarly, both variables also had high positive correlations with body dimensions, kilograms of bone mass, and lower limb perimeters. Moreover, peak power also showed a high positive correlation with kilograms of fat using Faulkner’s equation. It is important to note that, in both variables, there were no negative correlations with anthropometric variables related to fat in contrast to jump height.

Table 4, Table 5 and Table 6 show the descriptive statistics for CMJ post-WAnT, as well as the variation for absolute and relative values, providing insights into neuromuscular fatigue. Although no statistically significant differences were observed between positions, goalkeepers consistently showed the highest values across all variables, followed by forwards, defenders, and midfielders. This pattern aligns with CMJ pre-WAnT measurements.

In terms of variation, goalkeepers experienced the largest reductions in both jump height and power, which could indicate greater neuromuscular fatigue compared to other positions. Forwards also exhibited notable decreases, while defenders and midfielders showed smaller declines. Notably, midfielders experienced a more pronounced drop in peak power than jump height, suggesting that although they maintained jump height relatively well, their ability to produce peak power was more affected. This may be related to the specific demands of their position, which could emphasize repeated moderate-intensity actions over maximal power output.

These differences in the extent of the reductions may be linked to the neuromuscular recovery capacity of each position. However, the larger decreases observed in goalkeepers and forwards could also be attributed to their higher initial values of CMJ pre-WAnT, which might suggest a greater capacity for decline. Overall, the interaction between neuromuscular fatigue and initial performance levels seems to play a key role in the observed CMJ post-WAnT performance variations.

Regarding the correlation between anthropometric variables and CMJ post-WAnT data (Table 7), similar results to those observed in CMJ pre-WAnT. For jump height, there were moderate negative correlations with anthropometric variables related to fat, but in this case, a high negative correlation was specifically found with the suprailiac skinfold. For CMJ_Pmean_ and CMJ_Ppeak_, as with CMJ pre-WAnT, there was a very high positive correlation with body mass and kilograms of muscle, but only with Lee’s equation. Additionally, after WAnT, there was now a very high positive correlation with wingspan. CMJ_Pmean_ also showed a very high positive correlation with kilograms of bone mass post-WAnT. Other anthropometric variables with high positive correlations included body dimensions, bone diameters, and lower limb perimeters.

Table 8 shows the descriptive statistics for WAnT. Although no significant differences were found between positions for any of the variables, some trends are noteworthy. Goalkeepers consistently show the highest values across most variables, followed by forwards. An exception was found in P_min_ and forwards showed slightly higher values than goalkeepers. The TP_peak_ is similar across all positions.

Comparing midfielders and defenders, despite having similar overall values, midfielders have a higher P_peak_ and P_min_. In turn, defenders show a slightly higher P_mean_ in Split_20–30S_ and Split_15–30S_.

The high values of P_peak_ and P_mean_ observed for goalkeepers are consistent with observations from a study on Greek first-division players during the pre-season [21]. However, forwards have the second highest P_peak_ and P_mean_ values, whereas in the Greek study, midfielders held these values. Regarding FI, defenders and goalkeepers show the highest values, followed by forwards and midfielders, who have nearly identical values. Comparing the values between both studies, the data observed in this research are higher than those reported for Greek players. To the best of our knowledge, the study by Nikolaidis [21] is the only one to conduct a WAnT on professional female soccer players and differentiate by playing position.

It is important to note the lack of scientific literature conducting WAnT on female soccer players. Comparing the values from this research for all players, P_peak_ is lower (616.78 ± 84.42) than that observed in American collegiate players during the pre-season (716.87 ± 137.18) [47] but higher than that in Turkish (439.7 ± 61.1) [48] and Greek first-division players (578 ± 72) [49]. P_mean_ is greater (506.09 ± 55.52) compared to both American collegiate (356.61 ± 68.89) [47], Turkish [48] (316.1 ± 34.4), and Greek first-division players (431 ± 50) [49]. Additionally, FI is significantly lower than the values reported for the American (82.08 ± 11.86) [47], Turkish (55.7 ± 3.75) [48], and Greek players (45.9 ± 8.3) [49].

After examining the descriptive statistics by position, the focus shifts to understanding how WAnT performance variables correlate with anthropometric and body composition measurements (Table 9). P_peak_, defined as the highest mechanical strength achieved during any 5-s time period during the test, and an indicator of anaerobic power [48], shows an almost perfect positive correlation with body mass, and very high positive correlations with muscle mass (Lee’s equation), fat mass (Faulkner and Durnin equations), and perimeters such as thigh, contracted and relaxed arm, waist, and calf.

Moreover, P_mean_ is defined as the average strength produced during the test and an indicator of anaerobic capacity [48]. It shows a very high positive correlation with body mass, muscle mass (Lee’s equation), and wrist diameter.

To our knowledge, only one study correlates WAnT performance with anthropometric variables in female soccer players. This study on Greek first-division players [49] observed a moderate correlation of P_peak_ with body fat percentage, a high positive correlation with kilograms of fat mass, and a very high correlation with fat-free mass. For P_mean_, the Greek study found no significant correlation with body fat percentage, but there was a moderate positive correlation with kilograms of fat mass and a very high correlation with fat-free mass.

The descriptive statistics for Yo-Yo IR1 are presented in Table 10. All variables are included, though the most used measures for assessing aerobic capacity are the distance covered and the estimated VO_2max_ [15].

Regarding distance covered, goalkeepers covered the least distance. That fact agrees with the observations reported for the national teams of Montenegro [32], Serbia [50], and Chile [30], as well as Tunisian first-division [27] and South African semi-professional players [28]. As expected, goalkeepers recorded lower values in the distance covered by Yo-Yo IR1. This result aligns with their typical game performance, covering approximately 4000 m compared to field players who average around 10,000 m [51].

Following with distance covered, but analyzing the highest values, midfielders covered the most. This finding is concordant with observations in Tunisian first-division players [27] and the national team of Chile [30]. Other studies reported the most distance covered by defenders from South Africa [28] and the national team of Montenegro [32].

No significant differences were found between positions for the distance covered, aligning with observations in semi-professional South African players [28] and the Serbian national team [50]. Conversely, other studies have reported significant positional differences. For example, the Montenegro national team showed differences between defenders and both goalkeepers and forwards [32]. Similarly, the Chilean national team found differences between midfielders and all other field positions [30]. Furthermore, significant differences were observed between goalkeepers and the rest positions in Tunisian first-division players [27].

The distance covered reported in the present research for all players (1133 ± 375.52) is comparable to that observed in a 2013 study on German second-division players (1102 ± 316). However, the timing of this test within the season is not specified. As mentioned earlier, since the measurements were taken on the first day of the pre-season, this introduces a limitation in comparing these results with other studies due to seasonal variations in performance. This research shows a higher distance covered compared to Turkish first-division players (676.3 ± 156.4) [48] and the Montenegro national team at the end of the season (801.42 ± 253.09) [32]. However, these values are lower than those found in Danish first-division players (1379), Spanish first-division players during the season (1224 ± 255), and Brazilian first-division players at the end of the pre-season (1556.4 ± 470.23).

Regarding VO_2max_, there were no significant differences between positions. Midfielders had the highest VO_2max_ values, while goalkeepers had the lowest. These results are in line with those of the Chilean study [30], although the Chilean study found significant differences between these positions. The VO_2max_ values in this study are lower than those observed in the Chilean national team during the season [30]. When comparing total VO_2max_ values, the results of this research are similar to those reported for German second-division players [52]. In addition, these values are higher than those observed in Turkish first-division players [48] but lower than those observed in Spanish semi-professional players during the season [53].

These findings align with the physiological demands of each playing position. Midfielders typically cover the greatest distances during matches, requiring superior aerobic capacity to sustain both offensive and defensive contributions. Their ability to maintain high-intensity efforts throughout the game may explain their higher Yo-Yo IR1 performance compared to other positions.

After examining the descriptive statistics by position, the focus shifts to understanding how Yo-Yo IR1 performance variables correlate with anthropometric and body composition measurements (Table 11).

Regarding the distance covered, a moderate negative correlation was observed with the supraspinal skinfold. Other studies have also reported negative correlations with fat-related variables. For instance, a study on the Chilean national team players [30] found a moderate negative correlation between the distance covered and both body fat percentage and the sum of six skinfolds. Similarly, a study on Brazilian first-division players [8] reported a high negative correlation between the distance covered and the sum of six skinfolds.

For estimated VO_2max_, there was a moderate negative correlation with the supraspinal skinfold and wingspan. As with the distance covered, there is a lack of studies for comparison. Current literature indicates only one study has correlated anthropometric variables with estimated VO_2max_ based on distance covered by the Yo-Yo IR1. This study, conducted on semi-professional Spanish players [53], found a moderate negative correlation between BMI and estimated VO_2max_.

### Limitations

One limitation of this study was the relatively small sample size of 34 female players. Additionally, not all players were able to complete every test due to injuries or discomfort, which further reduced the number of participants in certain analyses. This impacted the statistical power of the findings, particularly for positions such as goalkeeper, with only two players per team available. However, it is important to note that access to professional soccer teams is inherently difficult, and these limitations are common in studies involving elite athletes. Future research with larger and more consistent samples is necessary to confirm and expand upon these results. Additionally, the lack of dual-energy X-ray absorptiometry (DEXA) assessment may have limited the precision of body composition measurements. However, its high cost and limited accessibility make it impractical for field-based studies in team sports. Instead, ISAK-standardized anthropometry was used, a validated and widely applied method in sports science. Future research integrating different assessment techniques could provide further insights.

## 4. Conclusions

This study highlights the significant influence of body composition on the physical performance of female soccer players at the onset of the pre-season immediately following the off-season. A strong positive correlation was observed between muscle and body mass with anaerobic power and capacity. On the other hand, fat mass was moderately negatively correlated with explosive strength, recovery capacity, and aerobic performance. Regarding positional differences, goalkeepers demonstrated the highest values in anaerobic power, as well as in explosive strength. In contrast, midfielders excelled in aerobic capacity. These findings emphasize the need for individualized training programs that focus on increasing muscle mass and managing fat levels to optimize both anaerobic and aerobic performance. In summary, muscle mass and body mass are strongly related to better anaerobic performance, reinforcing the importance of strength training in female players, while fat mass negatively impacts explosive strength, recovery capacity, and aerobic performance, emphasizing the need for an optimal body composition to enhance athletic performance. These insights are crucial for coaches and nutritionists in developing training strategies and personalized nutrition plans tailored to the needs of each position as players return from the off-season, aimed at enhancing the specific attributes that can optimize performance.

## Figures and Tables

**Figure 1 jfmk-10-00079-f001:**
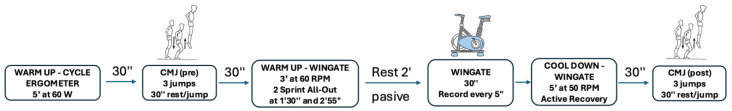
Timeline CMJ-WAnT.

**Table 1 jfmk-10-00079-t001:** Demographic data of the players: mean and standard deviations (mean ± SD).

	Mean ± SD
n	34
Age	23.06 ± 4.29
Height (cm)	164.15 ± 5.84
Weight (kg)	58.39 ± 6.62
BMI (kg/m^2^)	21.64 ± 21.59
∑6 Skinfolds (mm)	74.57 ± 18.48

BMI: Body Mass Index; ∑6 Skinfolds (triceps, subscapular, supraspinal, abdominal, front thigh, and medial calf).

**Table 2 jfmk-10-00079-t002:** Descriptive statistics of CMJ before WAnT: mean and standard deviations (mean ± SD).

Variable	MF	DF	FW	GK	Total	*p*-Value
Height_1_	24.46 ± 2.78	25.04 ± 3.31	26.97 ± 5.55	30.22 ± 5.71	26.01 ± 4.42	0.199
CMJ_Pmean1_	617.91 ± 54.43	622.56 ± 60.62	667.11 ± 97.78	757.60 ± 95.59	648.80 ± 83.97	0.046 *
CMJ_Ppeak1_	631.11 ± 54.34	633.38 ± 64.99	681.38 ± 101.27	773.59 ± 101.88	661.85 ± 87.31	0.049 *

MF: Midfielders; DF: Defenders; FW: Forwards; GK: Goalkeepers; Height: Jump height (cm); P_mean_: Mean power (W); P_peak_: Power peak (W). * Significant differences between positions (*p* < 0.05), as determined by ANOVA.

**Table 3 jfmk-10-00079-t003:** Correlation between anthropometry and CMJ before WAnT.

Anthropometry vs. CMJ Pre-WAnT	r	Magnitude	Lower 95% IC	Upper 95% IC	*p*-Value
Height					
Biceps (mm)	−0.448	Moderate	−0.692	−0.111	0.011
Muscle (%) (Poortmans)	0.444	Moderate	0.107	0.690	0.012
Fat (%) (Carter)	−0.429	Moderate	−0.680	−0.089	0.016
∑6 skinfolds	−0.429	Moderate	−0.680	−0.089	0.016
Fat (%) (Slaughter)	−0.415	Moderate	−0.671	−0.071	0.020
∑8 skinfolds	−0.410	Moderate	−0.667	−0.065	0.023
Fat (%) (Withers)	−0.398	Moderate	−0.659	−0.051	0.027
Fat (%) (Jackson & Pollock)	−0.390	Moderate	−0.654	−0.042	0.030
Endomorphy	−0.380	Moderate	−0.647	−0.029	0.035
Fat (kg) (Slaughter)	−0.370	Moderate	−0.641	−0.018	0.040
Fat (kg) (Withers)	−0.365	Moderate	−0.637	−0.012	0.045
CMJ_Pmean_					
Body Mass (kg)	0.772	Very High	0.575	0.884	<0.001
Muscle (kg) (Lee)	0.729	Very High	0.506	0.861	<0.001
Muscle (kg) (Poortmans)	0.704	Very High	0.466	0.847	<0.001
Wrist (bistyloid) (cm)	0.692	High	0.447	0.840	<0.001
Wingspan (cm)	0.679	High	0.427	0.833	<0.001
Bone Mass (kg) (Rocha)	0.674	High	0.420	0.830	<0.001
Height (cm)	0.649	High	0.383	0.816	<0.001
Calf (cm)	0.616	High	0.334	0.796	<0.001
Sitting Height (cm)	0.613	High	0.331	0.795	<0.001
CMJ_Ppeak_					
Body Mass (kg)	0.779	Very High	0.586	0.888	<0.001
Muscle (kg) (Lee)	0.733	Very High	0.511	0.863	<0.001
Muscle (kg) (Poortmans)	0.702	Very High	0.463	0.846	<0.001
Wrist (bistyloid) (cm)	0.683	High	0.434	0.835	<0.001
Wingspan (cm)	0.672	High	0.417	0.829	<0.001
Bone Mass (kg) (Rocha)	0.655	High	0.391	0.819	<0.001
Height (cm)	0.643	High	0.374	0.812	<0.001
Calf (cm)	0.634	High	0.361	0.807	<0.001
Fat (kg) (Faulkner)	0.596	High	0.306	0.785	<0.001
Sitting Height (cm)	0.594	High	0.304	0.784	<0.001
Humerus (cm)	0.575	High	0.277	0.772	<0.001

r = 0.0–0.09 were considered trivial, r = 0.10–0.29 small, r = 0.30–0.49 moderate, r = 0.50–0.69 high, r = 0.70–0.89 very high, r = 0.90–0.99 almost perfect, and r = 1 perfect correlation [45].

**Table 4 jfmk-10-00079-t004:** Descriptive statistics of CMJ after WAnT: mean and standard deviations (mean ± SD).

Variable	MF	DF	FW	GK	Total	*p*-Value
Height_2_	23.01 ± 3.20	23.50 ± 3.03	24.10 ± 5.74	26.47 ± 4.79	23.85 ± 4.18	0.680
CMJ_Pmean2_	598.40 ± 50.88	603.03 ± 58.26	630.68 ± 104.75	710.72 ± 102.19	621.17 ± 81.79	0.184
CMJ_Ppeak2_	609.75 ± 51.73	613.84 ± 56.66	645.71 ± 114.22	719.69 ± 106.51	633.31 ± 85.61	0.227

MF: Midfielders; DF: Defenders; FW: Forwards; GK: Goalkeepers; Height: Jump height (cm); P_mean_: Mean power (W); P_peak_: Power peak (W).

**Table 5 jfmk-10-00079-t005:** Variation of CMJ: mean and standard deviations (mean ± SD).

Variable	MF	DF	FW	GK	Total	*p*-Value
△Height	−1.45 ± 1.40	−1.54 ± 1.82	−2.87 ± 1.78	−3.76 ± 1.37	−2.16 ± 1.79	0.086
△CMJ_Pmean_	−19.51 ± 19.53	−19.53 ± 23.21	−36.44 ± 22.56	−46.87 ± 7.27	−27.63 ± 22.57	0.104
△CMJ_Ppeak_	−21.36 ± 23.87	−19.54 ± 32.17	−35.66 ± 26.53	−53.90 ± 4.64	−28.54 ± 27.89	0.196

MF: Midfielders; DF: Defenders; FW: Forwards; GK: Goalkeepers; P_mean_: Mean power (W); P_peak_: Power peak (W).

**Table 6 jfmk-10-00079-t006:** Variation of CMJ %: mean and standard deviations (mean ± SD).

Variable	MF	DF	FW	GK	Total	*p*-Value
△Height	−6.78 ± 7	−6.84 ± 7.90	−12.80 ± 8.45	−14.10 ± 4.39	−9.45 ± 7.93	0.188
△CMJ_Pmean_	−3.27 ± 3.35	−3.30 ± 3.82	−6.15 ± 3.98	−6.78 ± 2.05	−4.55 ± 3.79	0.185
△CMJ_Ppeak_	−3.54 ± 3.96	−3.21 ± 5.00	−6.05 ± 4.48	−7.67 ± 1.81	−4.65 ± 4.45	0.283

MF: Midfielders; DF: Defenders; FW: Forwards; GK: Goalkeepers; P_mean_: Mean power (W); P_peak_: Power peak (W).

**Table 7 jfmk-10-00079-t007:** Correlation between anthropometry and CMJ after WAnT.

Anthropometry vs. CMJ Post-WAnT	r	Magnitude	Lower 95% IC	Upper 95% IC	*p*-Value
Height					
Supraspinal (mm)	−0.502	High	−0.727	−0.180	0.004
Fat (kg) (Slaughter)	−0.483	Moderate	−0.715	−0.155	0.006
Fat (%) (Carter)	−0.484	Moderate	−0.716	−0.157	0.006
∑6 skinfolds	−0.484	Moderate	−0.716	−0.157	0.006
Fat (kg) (Withers)	−0.473	Moderate	−0.709	−0.143	0.008
∑8 skinfolds	−0.474	Moderate	−0.709	−0.144	0.008
Fat (%) (Slaughter)	−0.464	Moderate	−0.703	−0.131	0.009
Fat (%) (Withers)	−0.455	Moderate	−0.697	−0.120	0.010
Biceps (mm)	−0.452	Moderate	−0.695	−0.116	0.011
Fat (%) (Jackson & Pollock)	−0.449	Moderate	−0.693	−0.112	0.011
Fat (kg) (Carter)	−0.451	Moderate	−0.694	−0.115	0.012
Endomorphy	−0.439	Moderate	−0.687	−0.100	0.013
Muscle (%) (Poortmans)	0.429	Moderate	0.088	0.680	0.016
Medial Calf (mm)	−0.426	Moderate	−0.678	−0.084	0.017
Front thigh (mm)	−0.400	Moderate	−0.661	−0.054	0.026
Fat (%) (Durnin)	−0.390	Moderate	−0.654	−0.041	0.030
Fat (kg) (Durnin)	−0.390	Moderate	−0.654	−0.041	0.031
Fat (kg) (Jackson & Pollock)	−0.385	Moderate	−0.651	−0.036	0.033
Fat (%) (Faulkner)	−0.374	Moderate	−0.643	−0.023	0.038
CMJ_Pmean_					
Body Mass (kg)	0.760	Very High	0.554	0.878	<0.001
Muscle (kg) (Lee)	0.729	Very High	0.505	0.861	<0.001
Wingspan (cm)	0.711	Very High	0.476	0.851	<0.001
Bone Mass (kg) (Rocha)	0.706	Very High	0.469	0.848	<0.001
Muscle (kg) (Poortmans)	0.699	High	0.458	0.844	<0.001
Wrist (bistyloid) (cm)	0.687	High	0.440	0.838	<0.001
Height (cm)	0.675	High	0.421	0.831	<0.001
Sitting Height (cm)	0.652	High	0.388	0.818	<0.001
Calf (cm)	0.592	High	0.301	0.782	<0.001
CMJ_Ppeak_					
Body Mass (kg)	0.740	Very High	0.523	0.867	<0.001
Muscle (kg) (Lee)	0.722	Very High	0.494	0.857	<0.001
Wingspan (cm)	0.700	Very High	0.459	0.845	<0.001
Muscle (kg) (Poortmans)	0.697	High	0.456	0.843	<0.001
Bone Mass (kg) (Rocha)	0.686	High	0.439	0.837	<0.001
Wrist (bistyloid) (cm)	0.677	High	0.425	0.832	<0.001
Height (cm)	0.659	High	0.397	0.821	<0.001
Sitting Height (cm)	0.626	High	0.348	0.802	<0.001
Calf (cm)	0.588	High	0.295	0.780	<0.001

r = 0.0–0.09 were considered trivial, r = 0.10–0.29 small, r = 0.30–0.49 moderate, r = 0.50–0.69 high, r = 0.70–0.89 very high, r = 0.90–0.99 almost perfect, and r = 1 perfect correlation [45].

**Table 8 jfmk-10-00079-t008:** Descriptive statistics of WAnT: mean and standard deviations (mean ± SD).

Variable	MF	DF	FW	GK	Total	*p*-Value
P_peak_	600.25 ± 81.27	592.50 ± 64.95	633.64 ± 84.34	680.00 ± 145.59	616.78 ± 84.42	0.368
TP_peak_	7.00 ± 1.60	6.40 ± 1.35	6.64 ± 1.75	6.67 ± 3.22	6.66 ± 1.68	0.912
P_min_	385.25 ± 39.43	377.60 ± 37.89	399.64 ± 59.64	392.00 ± 35.55	388.44 ± 45.64	0.752
FI (%)	35.21 ± 7.56	35.91 ± 6.41	36.49 ± 9.58	41.29 ± 8.08	36.44 ± 7.83	0.727
P_mean_	491.99 ± 57.24	491.27 ± 41.69	520.87 ± 63.23	538.92 ± 61.53	506.09 ± 55.52	0.398
Split_0–10S_	550.44 ± 81.74	545.59 ± 53.99	585.60 ± 83.35	626.60 ± 130.63	568.15 ± 79.88	0.358
Split_10–20S_	517.77 ± 59.64	515.74 ± 53.49	546.68 ± 68.69	562.67 ± 41.53	531.28 ± 59.76	0.468
Split_20–30S_	418.11 ± 45.02	422.21 ± 40.24	441.22 ± 58.67	440.12 ± 27.14	429.40 ± 46.86	0.690
Split_0–15S_	547.87 ± 73.93	542.37 ± 50.28	580.95 ± 77.23	614.04 ± 101.34	563.72 ± 71.53	0.345
Split_15–30S_	441.20 ± 48.92	445.47 ± 44.51	466.14 ± 60.49	469.94 ± 28.87	453.80 ± 49.92	0.642

MF: Midfielders; DF: Defenders; FW: Forwards; GK: Goalkeepers; P_peak_: Peak power (W); TP_peak_: Time to P_peak_ (s); P_min_: Minimum power (W); FI (%): Fatigue index; P_mean_: Mean power (W); Split: P_mean_ calculated every 10 and 15 s.

**Table 9 jfmk-10-00079-t009:** Correlation between anthropometry and WAnT.

Anthropometry vs. WAnT	r	Magnitude	Lower 95% IC	Upper 95% IC	*p*-Value
P_peak_					
Body Mass (kg)	0.904	Almost perfect	0.810	0.952	<0.001
Muscle (kg) (Lee)	0.796	Very High	0.619	0.896	<0.001
Fat (kg) (Faulkner)	0.761	Very High	0.561	0.877	<0.001
Thigh (cm)	0.733	Very High	0.517	0.862	<0.001
Humerus (cm)	0.730	Very High	0.512	0.860	<0.001
Fat (kg) (Durnin)	0.730	Very High	0.512	0.860	<0.001
Relaxed arm (cm)	0.726	Very High	0.505	0.858	<0.001
Calf (cm)	0.726	Very High	0.505	0.857	<0.001
Contracted arm (cm)	0.713	Very High	0.485	0.850	<0.001
Waist (cm)	0.700	Very High	0.464	0.843	<0.001
Fat (kg) (Jackson & Pollock)	0.695	High	0.457	0.840	<0.001
Muscle (kg) (Poortmans)	0.680	High	0.435	0.832	<0.001
Wrist (bistyloid) (cm)	0.677	High	0.430	0.830	<0.001
Hip (cm)	0.666	High	0.414	0.824	<0.001
Sitting Height (cm)	0.654	High	0.395	0.816	<0.001
Bone Mass (kg) (Rocha)	0.642	High	0.378	0.810	<0.001
Fat (kg) (Carter)	0.631	High	0.362	0.803	<0.001
Height (cm)	0.629	High	0.359	0.802	<0.001
Hip (cm)	0.629	High	0.358	0.802	<0.001
Fat (kg) (Withers)	0.609	High	0.218	0.739	<0.001
Subscapular (mm)	0.567	High	0.273	0.765	<0.001
Wingspan (cm)	0.564	High	0.268	0.763	<0.001
P_mean_					
Body Mass (kg)	0.799	Very High	0.624	0.898	<0.001
Muscle (kg) (Lee)	0.763	Very High	0.565	0.878	<0.001
Wrist (bistyloid) (cm)	0.706	Very High	0.474	0.846	<0.001
Muscle (kg) (Poortmans)	0.677	High	0.430	0.830	<0.001
Bone Mass (kg) (Rocha)	0.667	High	0.415	0.824	<0.001
Sitting Height (cm)	0.659	High	0.403	0.820	<0.001
Height (cm)	0.649	High	0.388	0.813	<0.001
Humerus (cm)	0.625	High	0.353	0.799	<0.001
Fat (kg) (Faulkner)	0.619	High	0.345	0.796	<0.001
Wingspan (cm)	0.600	High	0.318	0.785	<0.001
Thigh (cm)	0.580	High	0.290	0.773	<0.001
Calf (cm)	0.580	High	0.290	0.772	<0.001
Split_0–10S_					
Body Mass (kg)	0.889	Very High	0.782	0.945	<0.001
Muscle (kg) (Lee)	0.796	Very High	0.619	0.896	<0.001
Fat (kg) (Faulkner)	0.777	Very High	0.587	0.886	<0.001
Fat (kg) (Durnin)	0.757	Very High	0.555	0.875	<0.001
Calf (cm)	0.751	Very High	0.546	0.872	<0.001
Thigh (cm)	0.747	Very High	0.538	0.869	<0.001
Humerus (cm)	0.724	Very High	0.503	0.857	<0.001
Fat (kg) (Jackson & Pollock)	0.711	Very High	0.482	0.849	<0.001
Relaxed arm (cm)	0.710	Very High	0.480	0.849	<0.001
Contracted arm (cm)	0.702	Very High	0.467	0.844	<0.001
Waist (cm)	0.686	High	0.443	0.835	<0.001
Muscle (kg) (Poortmans)	0.681	High	0.436	0.832	<0.001
Hip (cm)	0.669	High	0.418	0.825	<0.001
BMI (kg/m^2^)	0.662	High	0.408	0.821	<0.001
Fat (kg) (Carter)	0.645	High	0.382	0.811	<0.001
Fat (kg) (Withers)	0.630	High	0.360	0.802	<0.001
Wrist (bistyloid) (cm)	0.624	High	0.352	0.799	<0.001
Height (cm)	0.611	High	0.333	0.791	<0.001
Sitting Height (cm)	0.605	High	0.325	0.788	<0.001
Subscapular (mm)	0.589	High	0.303	0.778	<0.001
Bone Mass (kg) (Rocha)	0.575	High	0.283	0.770	<0.001
Split_10–20S_					
Body Mass (kg)	0.688	High	0.447	0.836	<0.001
Muscle (kg) (Lee)	0.679	High	0.433	0.831	<0.001
Wrist (bistyloid) (cm)	0.665	High	0.412	0.823	<0.001
Muscle (kg) (Poortmans)	0.633	High	0.364	0.804	<0.001
Bone Mass (kg) (Rocha)	0.589	High	0.303	0.778	<0.001
Humerus (cm)	0.583	High	0.293	0.774	<0.001
Fat (kg) (Faulkner)	0.570	High	0.277	0.767	<0.001
Height (cm)	0.564	High	0.269	0.763	<0.001
Sitting Height (cm)	0.560	High	0.263	0.760	<0.001
Split_20–30S_					
Bone Mass (kg) (Rocha)	0.631	High	0.362	0.803	<0.001
Wrist (bistyloid) (cm)	0.619	High	0.344	0.796	<0.001
Sitting Height (cm)	0.607	High	0.328	0.789	<0.001
Height (cm)	0.558	High	0.260	0.759	<0.001
Split_0–15S_					
Body Mass (kg)	0.877	Very High	0.762	0.939	<0.001
Muscle (kg) (Lee)	0.803	Very High	0.632	0.900	<0.001
Fat (kg) (Faulkner)	0.758	Very High	0.556	0.875	<0.001
Fat (kg) (Durnin)	0.722	Very High	0.498	0.855	<0.001
Thigh (cm)	0.719	Very High	0.494	0.853	<0.001
Calf (cm)	0.713	Very High	0.485	0.850	<0.001
Muscle (kg) (Poortmans)	0.703	Very High	0.469	0.845	<0.001
Humerus (cm)	0.699	High	0.463	0.842	<0.001
Relaxed arm (cm)	0.684	High	0.441	0.834	<0.001
Contracted arm (cm)	0.672	High	0.422	0.827	<0.001
Fat (kg) (Jackson & Pollock)	0.665	High	0.411	0.823	<0.001
Wrist (bistyloid) (cm)	0.663	High	0.408	0.821	<0.001
BMI (kg/m^2^)	0.646	High	0.383	0.812	<0.001
Waist (cm)	0.642	High	0.378	0.810	<0.001
Height (cm)	0.617	High	0.342	0.795	<0.001
Sitting Height (cm)	0.612	High	0.335	0.792	<0.001
Fat (kg) (Carter)	0.607	High	0.328	0.789	<0.001
Bone Mass (kg) (Rocha)	0.604	High	0.324	0.787	<0.001
Hip (cm)	0.603	High	0.322	0.786	<0.001
Fat (kg) (Withers)	0.587	High	0.300	0.777	<0.001
Wingspan (cm)	0.566	High	0.271	0.764	<0.001
Subscapular (mm)	0.560	High	0.262	0.760	<0.001
Split_15–30S_					
Wrist (bistyloid) (cm)	0.627	High	0.357	0.801	<0.001
Bone Mass (kg) (Rocha)	0.619	High	0.344	0.796	<0.001
Sitting Height (cm)	0.590	High	0.304	0.778	<0.001
Height (cm)	0.564	High	0.268	0.763	<0.001
Muscle (kg) (Lee)	0.563	High	0.266	0.762	<0.001

r = 0.0–0.09 were considered trivial, r = 0.10–0.29 small, r = 0.30–0.49 moderate, r = 0.50–0.69 high, r = 0.70–0.89 very high, r = 0.90–0.99 almost perfect, and r = 1 perfect correlation [45].

**Table 10 jfmk-10-00079-t010:** Descriptive statistics of test Yo-Yo IR1: mean and standard deviations (mean ± SD).

Variable	MF	DF	FW	GK	Total	*p*-Value
Speed Level	16.75 ± 1.12	16.11 ± 1.15	15.72 ± 0.65	14.30 ± 1.41	16.03 ± 1.20	0.068
Speed (km/h)	14.84 ± 0.45	14.63 ± 0.40	14.54 ± 0.20	14.06 ± 0.48	14.62 ± 0.42	0.137
Distance (m)	1366.67 ± 355.45	1155.00 ± 376.11	1016.00 ± 221.99	640.00 ± 339.41	1133.33 ± 375.52	0.086
Time (s)	685.33 ± 174.13	580.75 ± 184.84	512.60 ± 110.31	323.50 ± 171.83	569.91 ± 185.43	0.083
VO_2max_	47.88 ± 2.99	46.10 ± 3.16	44.93 ± 1.87	41.78 ± 2.85	45.92 ± 3.15	0.086

MF: Midfielders; DF: Defenders; FW: Forwards; GK: Goalkeepers.

**Table 11 jfmk-10-00079-t011:** Correlation between anthropometry and Yo-Yo IR1.

Anthropometry vs. Yo-Yo IR1	r	Magnitude	Lower 95% IC	Upper 95% IC	*p*-Value
Distance					
Supraspinal (mm)	−0.452	Moderate	−0.739	−0.025	0.040
VO_2max_					
Supraspinal (mm)	−0.452	Moderate	−0.739	−0.025	0.040
Wingspan (cm)	−0.454	Moderate	−0.740	−0.027	0.039

r = 0.0–0.09 were considered trivial, r = 0.10–0.29 small, r = 0.30–0.49 moderate, r = 0.50–0.69 high, r = 0.70–0.89 very high, r = 0.90–0.99 almost perfect, and r = 1 perfect correlation [45].

## Data Availability

The data are contained within the article.

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
