# Peer review of "Relationship Between Anthropometric Profile, Body Composition, and Physical Performance in Spanish Professional Female Soccer Players at Pre-Season Onset: A Cross-Sectional Study"

_jfmk, 2025, doi:10.3390/jfmk10010079_

Round 1
Reviewer 1 Report
Comments and Suggestions for Authors
Dear colleagues, Thank you for the opportunity to read and contribute to the manuscript. Please find attached recommendations and suggestions for improving the manuscript.
Sincerely

The English language must be revised.
Reviewer 2 Report
Comments and Suggestions for Authors
Dear Authors,
Congratulations on the research you conducted; however, it must be accompanied by a proper statistical analysis. Below, you will find my detailed comments.
Abstract
The abstract is concise and informative, clearly describing the study's purpose, methods, key results, and conclusions.
Introduction
The introduction is comprehensive, scientifically sound (with references where necessary), and presents the theoretical framework of the study clearly. However, it could be improved by better clarifying the research gaps. The authors could be more specific about what is already known (using the necessary references) and what remains unexplored.
Materials and Methods
The section is well-structured, and the descriptions of the tests and anthropometric measurements are thorough and follow established methodologies.
Results and Discussion
This section is extensive and includes 11 tables, but there is no mention of the number of players per position or division (first or second).
Additionally, there is no indication of whether the data follow a normal distribution for all groups (player positions), which is a prerequisite for performing a One-way ANOVA. Given that the goalkeepers are very few (as acknowledged by the authors in the study limitations), it is likely that normality is not met for this specific group. Furthermore, effect sizes are not reported in the One-way ANOVA analysis.
Another important aspect not considered is that the study includes players from both the first and second divisions, who may differ in performance levels. If the proportion of first- and second-division players varies across positions, significant biases could arise. To account for this, the authors should have used a Two-way ANOVA or a regression analysis instead of a One-way ANOVA.
Moreover, simple correlations do not account for mediators and confounders, which could lead to misleading conclusions. For example, the correlation between players' power and total body mass may actually be due to these players having higher muscle mass. Conducting such a large number of statistical tests also increases the risk of Type I errors. Therefore, instead of multiple correlation analyses, regression analyses should have been performed to control for potential confounders.
Overall, this section is difficult for a reader to follow, and the statistical analyses require revision.
Conclusions
The conclusions are well-written, clear, and impactful, with strong practical implications.
Author Response
Por favor vea el archivo adjunto.

Reviewer 3 Report
Comments and Suggestions for Authors
The manuscript entitled “Relationship Between Anthropometric Profile, Body Composition, and Physical Performance in Spanish Professional Female Soccer Players at Pre-Season Onset” was evaluated. The article presents important information about female soccer players; however, adjustments need to be made so that it can be published in this journal.
Below are some suggestions for analysis and improvements in the work:
Abstract
1- Please mention in the abstract which statistical test you used, did the data present a normal distribution or not?
2- In the results, it is important to present the p value together with the r value. To make it easier for the reader, present the p and r values in parentheses right after mentioning what was correlated, for example, “Very high positive correlations were found 26 between body and muscle mass with the power variables in CMJ (r=xxxx; p= xxx) and WAnT (r = xxx; p=xxxx). Adjustment throughout the abstract.
Keywords
3- Avoid repeating words presented in the title, use similar terms to increase the search capacity of your manuscript by interested readers.
Introduction
4- In the introduction, you mention that there are few studies involving female soccer players, especially in relation to the analyses that will be addressed in this study. However, you mention that studies with men are widely disseminated. Therefore, I believe that you could present in the introduction the findings on male soccer players regarding the correlations between anthropometry and body composition and the outcome variables of this study. With this, you can provide the reader with some information about what can be expected in the work. Include studies with gold standard techniques in relation to body composition and studies with the evaluation of skinfolds to see if they follow the same outcome, since you worked with skinfolds.
5- The end of the introduction presents the primary and secondary objectives. In relation to the secondary objective, please check what is presented in the abstract and standardize the writing.
Materials and Methods
6- Between lines 78 and 80, there is no need to repeat the objective of the study.
7- In line 87, please replace the website with the registry link, this will make it easier for the reader to consult the registry information.
8- I couldn't find anything in the methodology about the sample calculation. Did you, do it?
9- In the inclusion and exclusion criteria, it was not stated whether there was a minimum or maximum age to participate in the study. From what I observed, there were athletes whose parents had to sign the terms. How old would these athletes be?
10- To make it easier for the reader to understand how many people were invited to participate in the research and how many were excluded during the process, please include in the work a flowchart adapted according to CONSORT (https://view.officeapps.live.com/op/view.aspx?src=https%3A%2F%2Fwww.equator-network.org%2Fwp-content%2Fuploads%2F2013%2F09%2FCONSORT-2010-Flow-Diagram-MS-Word.doc&wdOrigin=BROWSELINK).
11- Between lines 106 and 108, even though you mention that the methodology and procedures were reported in previous work, you must describe the basic information about the equipment, which were the eight skinfolds and the methodology for calculating body composition from them. I would also like to know if you did not evaluate body composition in other ways, such as DEXA or bioimpedance. This information would be important to better understand body composition and, consequently, strengthen the findings.
12- In Figure 1, I believe it is important to create a caption or explain in the text how long the warm-up was before starting the tests. It was also not clear what the difference was between the first warm-up (before pre-CJM) and the second (before the post-CMJ). Is this protocol presented in Figure 1 adapted from an article or was it developed by you?
13- In the analysis, it was not observed whether you performed normality tests on the samples to determine whether the data were parametric or not. This issue is important to raise, since nonparametric data should not be presented as mean and standard deviation, but rather as median and interquartile range, for example. And, since you perform parametric and nonparametric correlation tests, it is assumed that there were variables with normal and nonnormal patterns. Please check and adjust.
Results and Discussion
14- In Table 1, the abbreviation BMI is observed; however, the text does not show the meaning of the term in full. Check and adjust, if possible, describing all abbreviations as captions in the tables, even if they have already been presented in the text.
15- In the caption of Table 2, explain more clearly that the p value is from the post hoc test, which shows that the goalkeepers had a better performance than the other athletes. As it is, it is not clear whether the p is related to the ANOVA or the post hoc. Adjust this information for similar tables.
16- In Table 3, you only provided the information that showed correlation. Please also include the values of those that did not show significant correlation so that the reader can have a broader view of the study. Adjust this information for similar tables.
17- In Table 10, I did not understand the asterisk, since the p value is 0.068.
18- Regarding the discussion, I missed a more biological explanation of the findings, for example, how adipocytes or muscle cells can influence the performance of athletes. Explain a little about the bioenergetics of football to try to justify the anaerobic tests showing more correlation in relation to the cardiorespiratory capacity of athletes.
Limitations
19- I believe that a major limitation of the study was the fact that you did not use a gold standard such as DEXA to assess the athletes' body composition or perhaps use bioimpedance. I say this because, since the focus of the study was body composition, more precise techniques should be applied to strengthen the results. For example, the fact that a certain skin fold shows correlation does not represent the body as a whole.
20- You could also have assessed VO2max directly to have greater robustness in the statistical analyses.
Reviewer 4 Report
Comments and Suggestions for Authors
Title and Abstract
Title:
Consider simplifying and enhancing the clarity for a broader audience. Suggested Title: "Anthropometric Profile, Body Composition, and Physical Performance in Professional Female Soccer Players at Pre-Season Onset: A Cross-Sectional Study."
Abstract:
Background: Improve flow and emphasize the knowledge gap.
"Body composition is a key factor in both anaerobic and aerobic performance in professional soccer. However, the relationship between body composition and performance metrics in professional female players remains underexplored, particularly at pre-season onset."
Results: Mention positional differences more clearly in the abstract.
"Goalkeepers displayed higher anaerobic power and jump performance than outfield players (p < 0.05)."
Conclusions: Reinforce the application of findings.
"Monitoring anthropometric and body composition profiles is essential to optimize performance in female soccer players, especially at pre-season onset."
Introduction
Strengthen the Research Gap:
Could you mention the scarcity of data on female soccer players compared to males, especially regarding pre-season assessments?
Highlight the increasing professionalization of women’s soccer and the need for tailored training strategies.
Refinement of Key Sentences:
“Soccer is an intense multidirectional and intermittent sport…” → “Soccer is a multidirectional, high-intensity intermittent sport where success depends on physical, technical, tactical, and anthropometric factors.”
Materials and Methods
Participants:
For international readers, clarify that the teams belonged to the Spanish Primera División and Segunda División.
Mention that parental consent was obtained for underage players more explicitly.
Anthropometric Measurements:
Briefly describe the purpose of skinfolds, breadths, and perimeters (e.g., to estimate body composition and somatotype).
Consider adding a table summarizing measurement protocols and tools (e.g., Harpenden caliper, SECA scale).
Performance Tests:
Could you include the reliability (e.g., intra- and inter-rater reliability) or cite previous studies validating CMJ, WAnT, and Yo-Yo IR1 for female athletes?
Clarify why Yo-Yo IR1 was conducted two days later—mention potential recovery considerations.
Results
Presentation of Data:
Integrate p-values and effect sizes (e.g., Cohen’s d) into tables highlighting positional differences.
Indicate the number of participants completing each test (some tests had dropouts).
Graphical representation (e.g., bar plots with confidence intervals) could enhance the visualization of positional differences.
Key Clarifications:
CMJ pre- and post-WAnT differences could be better contextualized by emphasizing neuromuscular fatigue implications.
Reinforce that goalkeepers consistently exhibited higher power outputs due to their positional demands.
Please ensure that midfielders had the highest Yo-Yo IR1 performance, aligning with their aerobic demands.
Discussion
Key Improvements:
Link Results to Practical Applications: Relate the findings to training practices (e.g., goalkeepers may benefit from power-based training and midfielders from aerobic conditioning).
Contextualize Within Literature: Compare your results with studies on elite male players, highlighting similarities and differences in positional demands.
Seasonal Variability: Discuss the impact of pre-season timing on performance and body composition (e.g., higher fat levels or reduced aerobic capacity post-off-season).
Specific Revisions:
“This may be related to their role-specific demands…” → “Goalkeepers' elevated power outputs likely reflect the explosive and jump-oriented demands of their position.”
“Muscle and body mass are strongly related to anaerobic performance…” → “Muscle and body mass positively influence anaerobic performance, emphasizing the importance of strength training in female players.”
Limitations
Could you highlight that the small sample size limits the generalizability of positional comparisons?
Mention the lack of longitudinal data to assess changes across the season.
Could you consider adding a note on the potential influence of menstrual cycle phases on performance, as this is increasingly recognized in female athlete research?
Conclusion
Refine for impact:
“Anthropometric and body composition profiles significantly influence the physical performance of professional female soccer players. Position-specific demands further shape performance characteristics, underscoring the need for individualized conditioning strategies at the pre-season onset.”
Tables and Figures
Ensure uniform formatting (e.g., consistent decimal places and units).
Could you add a summary table in the discussion consolidating positional differences and their practical implications?
References
Review in-text citations for consistency with the reference list.
Update key references, especially regarding female-specific soccer research, if newer studies (2023–2024) are available.
The English could be improved to express the research more clearly.
Round 2
Reviewer 1 Report
Comments and Suggestions for Authors
The manuscript “Relationship Between Anthropometric Profile, Body Composition, and Physical Performance in Spanish Professional Female Soccer Players at Pre-Season Onset: A cross-sectional study” has been resubmitted, and we have verified that most of the requests have been met or duly justified in a plausible manner.
After this review, we have no further contributions to make, as the manuscript now meets the necessary criteria for publication.
We would like to thank the authors for their adjustments and attention to the suggestions made in the previous round.
We note that at this stage there has been a greater concern for the English language, however, it still needs to be improved to ensure greater clarity and fluidity in reading. We recommend that the manuscript undergo a professional linguistic review before being published.
Reviewer 2 Report
Comments and Suggestions for Authors
Dear authors,
None of my concerns regarding the statistical analyses have been addressed. The simple statistical tests you have used are not suitable for such a large number of variables, as they do not account for confounders. In these cases, the use of multivariable models is essential.
Reviewer 3 Report
Comments and Suggestions for Authors
Dear Editor and Authors,
After reviewing the changes made to the manuscript by the authors, I believe that my comments have been addressed satisfactorily. Therefore, I believe that the manuscript can be accepted for publication.
Reviewer 4 Report
Comments and Suggestions for Authors
With the changes made, the article can be published.